# Geometric Random Walk Graph Neural Networks via Implicit Layers

## Abstract

Graph neural networks have recently attracted a lot of attention and have been applied with great success to several important graph problems. The Random Walk Graph Neural Network model was recently proposed as a more intuitive alternative to the well-studied family of message passing neural networks. This model compares each input graph against a set of latent "hidden graphs" using a kernel that counts common random walks up to some length. In this paper, we propose a new architecture, called Geometric Random Walk Graph Neural Network (GRWNN), that generalizes the above model such that it can count common walks of infinite length in two graphs. The proposed model retains the transparency of Random Walk Graph Neural Networks since its first layer also consists of a number of trainable "hidden graphs" which are compared against the input graphs using the geometric random walk kernel. To compute the kernel, we employ a fixed-point iteration approach involving implicitly defined operations. Then, we capitalize on implicit differentiation to derive an efficient training scheme which requires only constant memory, regardless of the number of fixed-point iterations. The employed random walk kernel is differentiable, and therefore, the proposed model is end-to-end trainable. Experiments on standard graph classification datasets demonstrate the effectiveness of the proposed approach in comparison with state-of-the-art methods.

## 1 Introduction

Recent years have witnessed an enormous growth in the amount of data represented as graphs. Indeed, graphs emerge naturally in several domains, including social networks, bioinformatics, and neuroscience, just to name a few. Besides the increase in the amount of graph-structured data, there is also a growing interest in applying machine learning techniques to data modeled as graphs. Among others, the graph classification and graph regression tasks have attracted a great deal of attention in the past years. These tasks have served as the fundamental building block within applications that deal with problems ranging from drug design (Kearnes et al., 2016) to session-based recommendation (Wu et al., 2019).

Graph Neural Networks (GNNs) provide a powerful tool for machine learning on graphs, So far, the field of GNNs has been largely dominated by message passing architectures. Indeed, most of them share the same basic idea, and can be reformulated into a single common framework, so-called message passing neural networks (MPNNs) (Gilmer et al., 2017). These models employ a message passing procedure to aggregate local information of vertices. For graph-related tasks, MPNNs usually apply some permutation invariant readout function to the vertex representations to produce a representation for the entire graph. The family of MPNNs has been heavily studied in the past few years, and there are now available very expressive models which have achieved state-of-the-art results in several tasks (Xu et al., 2019; Morris et al., 2019). Although the family of MPNNs is perhaps the most successful story in the field of graph representation learning, there exist models that follow different design paradigms and do not fall into this family. An example of such a model is the recently proposed Random Walk Graph Neural Network (RWNN) (Nikolentzos & Vazirgiannis, 2020). This model contains a number of trainable "hidden graphs", and it compares the input graphs against these graphs using a random walk kernel which counts the number of common walks in two graphs. The emerging kernel values are fed into a fully-connected neural network which acts as the classifier or regressor.

The employed random walk kernel is differentiable, and thus RWNN is end-to-end trainable. However, this kernel considers only random walks of a small length. Such local patterns may fail to capture the overall large-scale shape of the graphs, while several interesting properties of graphs depend on the graph's global structure. Furthermore, increasing the length of the walks has a direct impact on the model's computational complexity.

In this paper, we propose a novel approach to tackle these challenges. Specifically, we propose a new architecture, called Geometric Random Walk Graph Neural Network (GRWNN), that generalizes the RWNN model such that it can count common walks of infinite length in two graphs. The model contains a number of trainable "hidden graphs", and it compares the input graphs against these graphs using the geometric random walk kernel. Thus, instead of walks of small length, the proposed model considers walks of infinite length. To compute the kernel, GRWNN uses a fixed-point iteration approach. The kernel values are then passed on to a fully-connected neural network which produces the output. The proposed neural network is end-to-end trainable since we can directly differentiate through the fixed-point equations via implicit differentation, which leads to a very efficient implementation in terms of memory requirements. Hence, we can still update the "hidden graphs" during training with backpropagation. We compare the performance of the proposed model to state-of-the-art graph kernels and recently-proposed neural architectures on several graph classification datasets. Results show that in most cases, the GRWNN model matches or outperforms competing methods. Our main contributions are summarized as follows:

- We propose a novel neural network model, Geometric Random Walk Graph Neural Network, which employs the geometric random walk kernel to produce graph representations. The model counts common walks of infinite length in the input graph and a set of randomly initialized "hidden graphs".

- We employ an efficient scheme to compute the random walk graph kernel using fixed-point iterations. We show that we can directly differentiate through the fixed-point equations via implicit differentation, which leads to an efficient implementation.

- We evaluate the model's performance on several standard graph classification datasets and show that it achieves results similar and in some cases superior to those obtained by recent GNNs and graph kernels.

The rest of this paper is organized as follows. Section 2 provides an overview of the related work. Section 3 introduces some preliminary concepts. Section 4 provides a detailed description of the proposed model. Section 5 evaluates the proposed model in graph classification tasks. Finally, Section 6 concludes.

## 2   Related Work

Graph kernels have a long history in the field of graph representation learning (Kriege et al., 2020). A graph kernel is a kernel function between graphs, i.e., a symmetric positive semidefinite function defined on the space of graphs. These methods generate implicitly (or explicitly) graph representations and enable the application of kernel methods such as the SVM classifier to graphs. Most graph kernels are instances of the $R$-convolution framework (Haussler, 1999), and they compare substructures extracted from the graphs to each other. Such substructures include shortest paths (Borgwardt & Kriegel, 2005), random walks (Gärtner et al., 2003; Kashima et al., 2003), small subgraphs (Shervashidze et al., 2009), and others. Our work is related to random walk kernels, i.e., kernels that compare random walks to each other. The first such kernels were proposed by Gärtner et al. (2003) and by Kashima et al. (2003). The work of Kashima et al. was later refined by Mahé et al. (2004). Vishwanathan et al. (2010) and Kang et al. (2012) proposed new algorithms for efficiently computing random walk kernels. These algorithms improve the time complexity of kernel computation. Sugiyama & Borgwardt (2015) studied the problem of halting (i.e., longer walks are downweighted so much that the kernel value is completely dominated by the comparison of walks of length 1) that occurs in random walk kernels, and showed that its extent depends on properties of the graphs being compared. Zhang et al. (2018b) defined a different kernel which does not compare random walks to each other, but instead, compares the return probabilities of random walks. Finally, Kalofolias

et al. (2021) proposed a variant of the random walk kernel where structurally dissimilar vertices are not just down-weighed, but are not allowed to be visited during the simultaneous walk.

Although the first GNNs were proposed several years ago (Sperduti & Starita, 1997; Scarselli et al., 2009; Micheli, 2009), until recently, these models had attracted limited attention. In recent years, with the rise of deep learning, a lot of models started to emerge (Bruna et al., 2014; Li et al., 2015; Duvenaud et al., 2015; Atwood & Towsley, 2016; Defferrard et al., 2016; Lei et al., 2017). Most models update the representation of each vertex by aggregating the feature vectors of its neighbors. This update procedure can be viewed as a form of message passing algorithm and thus, these models are known as message passing neural networks (MPNNs) (Gilmer et al., 2017). To compute a feature vector for the entire graph, MPNNs apply some permutation invariant readout function to all the vertices of the graph. The family of MPNNs has been heavily studied in the past few years, and there are now available several sophisticated models which can produce expressive graph representations (Xu et al., 2019; Morris et al., 2019; Dehmamy et al., 2019; Morris et al., 2020). Despite the general recent focus on MPNNs, some works have proposed architectures that are not variants of this family of models (Niepert et al., 2016; Maron et al., 2019b;a; Nikolentzos & Vazirgiannis, 2020). The work closest to ours is the one reported in Nikolentzos & Vazirgiannis (2020) which presents the Random Walk Graph Neural Network (RWNN) model. In fact, in this paper, we generalize the RWNN model to compare random walks of infinite length in two graphs. Recently, another method that uses random walks to extract features which are then processed by a standard convolutional neural network was proposed (Toenshoff et al., 2021). However, the proposed approach decouples data representation from learning since random walks are sampled in a preprocessing stage. Our work is also related to implicit models which have been applied successfully to many problems (de Avila Belbute-Peres et al., 2018; Chen et al., 2018; Amos et al., 2018; Bai et al., 2019). The outputs of these models are determined implicitly by a solution of some underlying sub-problem. Implicit models have also been defined in the context of graph representation learning. For instance, Gu et al. (2020) proposed IGNN, a model that seeks the fixed-point of some equation which is equivalent to running an infinite number of message passing iterations. Thus, the final representation potentially contains information from all neighbors in the graph capturing long-range dependencies. Gallicchio & Micheli (2020) proposed a similar model which generates graph representations based on the fixed point of a recursive/dynamical system, but is actually only partially trained. In contrast to these approaches whose objective is to apply a large (or infinite) number of message passing layers implicitly, in our setting, we employ a fixed-point iteration approach to compute the random walk kernel and then we directly differentiate through the fixed point equations via implicit differentation.

## 3 Preliminaries

In this section, we begin by introducing our notation, and we then review the definition of the geometric random walk kernel.

### 3.1 Notation

Let $[n] = \{1, \ldots, n\} \subset \mathbb{N}$ for $n \geq 1$. Let $G = (V, E)$ be an undirected graph, where $V$ is the vertex set and $E$ is the edge set. We will denote by $n$ the number of vertices and by $m$ the number of edges. The neighbourhood $\mathcal{N}(v)$ of a vertex $v$ is the set of all vertices adjacent to $v$. Hence, $\mathcal{N}(v) = \{u \mid (v, u) \in E\}$ where $(v, u)$ is an edge between vertices $v$ and $u$ of $V$. The adjacency matrix $\mathbf{A} \in \mathbb{R}^{n \times n}$ of a graph $G$ is a symmetric (typically sparse) matrix used to encode edge information in the graph. The element of the $i^{th}$ row and $j^{th}$ column is equal to the weight of the edge between vertices $v_i$ and $v_j$ if such an edge exists, and 0 otherwise. The degree $d(v)$ of a vertex $v$ is equal to the sum of the weights of the edges that are adjacent to the vertex. For vertex-attributed graphs, every vertex in the graph is associated with a feature vector. We use $\mathbf{X} \in \mathbb{R}^{n \times d}$ to denote the vertex features where $d$ is the feature dimensionality. The feature of a given vertex $v_i$ corresponds to the $i^{th}$ row of $\mathbf{X}$.

The direct (tensor) product $G_\times = (V_\times, E_\times)$ of two graphs $G = (V, E)$ and $G' = (V', E')$ is defined as follows:

$$V_\times = \{(v, v') \in V \times V'\}$$
$$E_\times = \{((v, v'), (u, u')) \in V_\times \times V_\times \mid (v, u) \in E, \text{ and } (v', u') \in E'\}$$

We denote by $\mathbf{A}_\times$ the adjacency matrix of $G_\times$, and denote by $\Delta_\times$ and $\bar{d}_\times$ the maximum and average of the vertex degrees of $G_\times$, respectively. Thus, $\bar{d}_\times = 1/n \sum_{v \in V_\times} d(v)$. A walk in a graph is a sequence of vertices such that consecutive vertices are linked by an edge. Performing a random walk on the direct product $G_\times$ of two graphs $G$ and $G'$ is equivalent to performing a simultaneous random walk on the two graphs $G$ and $G'$.

We use $\otimes$ to represent the Kronecker product, and use $\odot$ to represent elementwise multiplication between two matrices or vectors of the same dimension. For a $p \times q$ matrix $\mathbf{V}$, $\text{vec}(\mathbf{V}) \in \mathbb{R}^{pq}$ represents the vectorized form of $\mathbf{V}$, obtained by stacking its columns. Let also $\text{vec}^{-1}$ denote the inverse vectorization operator which transforms a vector into a matrix, i.e., for a $pq$ vector $\mathbf{v}$, $\mathbf{V} = \text{vec}^{-1}(\mathbf{v})$ where $\mathbf{V} \in \mathbb{R}^{p \times q}$ (see the appendix for the exact definition of the vec and $\text{vec}^{-1}$ operators).

### 3.2 Random Walk Kernel

Given two graphs $G$ and $G'$, the random walk kernel counts all pairs of matching walks on $G$ and $G'$ (Gärtner et al., 2003). There are different variants of the kernel. For instance, the $p$-step random walk kernel (where $p \in \mathbb{N}$) counts all pairs of matching walks up to length $p$ on two graphs. The number of matching walks can be obtained through the adjacency matrix $\mathbf{A}_\times$ of the product graph $G_\times$ (Vishwanathan et al., 2010) since a random walk on $G_\times$ is equivalent to a simultaneous random walk on the two graphs. Assuming a uniform distribution for the starting and stopping probabilities over the vertices of two graphs, the $p$-step random walk kernel is defined as:

$$\kappa^p(G, G') = \sum_{i=1}^{|V_\times|} \sum_{i=j}^{|V_\times|} \left[ \sum_{l=0}^{p} \lambda_l \mathbf{A}_\times^l \right]_{ij}$$

where $\lambda_0, \lambda_1, \lambda_2, \ldots, \lambda_p$ are positive, real-valued weights, and $\mathbf{A}_\times^0$ is the identity matrix, i.e., $\mathbf{A}_\times^0 = \mathbf{I}$. For $p \to \infty$, we obtain $\kappa^\infty(G, G')$ which is known as the random walk kernel.

It turns out that if the sequence of weights $\lambda_0, \lambda_1, \lambda_2, \ldots$ coresponds to the geometric sequence defined as $\lambda_l = \lambda^l$, then the limit $\kappa^\infty(G, G')$ can be computed analytically as follows:

$$k^\infty(G, G') = \sum_{i=1}^{|V_\times|} \sum_{i=j}^{|V_\times|} \left[ \sum_{l=0}^{\infty} \lambda^l \mathbf{A}_\times^l \right]_{ij} = \sum_{i=1}^{|V_\times|} \sum_{i=j}^{|V_\times|} \left[ (\mathbf{I} - \lambda \mathbf{A}_\times)^{-1} \right]_{ij} = \mathbf{1}^\top (\mathbf{I} - \lambda \mathbf{A}_\times)^{-1} \mathbf{1} \tag{1}$$

It is well-known that the geometric series of matrices $\mathbf{I} + \lambda \mathbf{A}_\times + (\lambda \mathbf{A}_\times)^2 + \ldots$ converges only if the the largest-magnitude eigenvalue of $\mathbf{A}_\times$ (which is also the maximum eigenvalue if $G_\times$ is a graph with non-negative edge weights), denoted by $\mu_\times^{max}$, is strictly smaller than $1/\lambda$. Therefore, the geometric random walk kernel $k^\infty$ is well-defined only if $\lambda < 1/\mu_\times^{max}$. Interestingly, the maximum eigenvalue of $\mathbf{A}_\times$ is sandwiched between the average and the maximum of the vertex degrees of $G_\times$ (Brouwer & Haemers, 2011). We thus have that $\bar{d}_\times \leq \mu_\times^{max} \leq \Delta_\times$, and by setting $\lambda < 1/\Delta_\times$, the geometric series of matrices is guaranteed to converge.

By defining initial and stopping probability distributions over the vertices of $G$ and $G'$, we can obtain a probabilistic variant of the geometric random walk kernel. Let $\mathbf{p}$ and $\mathbf{p}'$ be two vectors that represent the initial probability distributions over the vertices of $G$ and $G'$. Likewise, let $\mathbf{q}$ and $\mathbf{q}'$ denote stopping probability distributions over the vertices of $G$ and $G'$. For uniform distributions for the initial and stopping probabilities over the vertices of the two graphs, we have $\mathbf{p}_i = \mathbf{q}_i = 1/|V|$ and $\mathbf{p}'_i = \mathbf{q}'_i = 1/|V'|$. Then, $\mathbf{p}_\times = \mathbf{p}\,\mathbf{p}'^\top$ and $\mathbf{q}_\times = \mathbf{q}\,\mathbf{q}'^\top$, and the variant of the geometric random walk kernel can be computed as $k^\infty(G, G') = \text{vec}(\mathbf{q}_\times)^\top (\mathbf{I} - \lambda \mathbf{A}_\times)^{-1} \text{vec}(\mathbf{p}_\times)$.

## 4 Geometric Random Walk Graph Neural Networks

The proposed GRWNN model maps input graphs to vectors by comparing them against a number of "hidden graphs", i.e., graphs whose adjacency and attribute matrices are trainable. The function that we employ to compare the input graphs against the "hidden graphs" is the geometric random walk graph kernel, one of the most well-studied kernels between graphs (Gärtner et al., 2003; Mahé et al., 2004; Vishwanathan et al., 2010). The proposed GRWNN model contains $N$ "hidden graphs" in total. The graphs may differ from each

other in terms of size (i.e., number of vertices). Furthermore, the vertices and/or edges of those graphs can be annotated with continuous multi-dimensional features. As mentioned above, both the structure and the vertex attributes (if any) of these "hidden graphs" are trainable. Thus, the adjacency matrix of a "hidden graph" $G_i$ of size $n$ is described by a trainable matrix $\mathbf{W}_i \in \mathbb{R}^{n \times n}$, while the vertex attributes are contained in the rows of another trainable matrix $\mathbf{Q}_i \in \mathbb{R}^{n \times d}$. Note that the "hidden graphs" correspond to weighted graphs, which can be directed or undirected graphs with or without self-loops. In our implementation, we constraint them to be undirected graphs without self-loops ($n(n-1)/2$ trainable parameters in total).

To compare an input graph $G$ against a "hidden graph" $G_i$, the model uses the geometric random walk kernel that was introduced in the previous section:

$$k^\infty(G, G_i) = \sum_{i=1}^{|V_\times|} \sum_{i=j}^{|V_\times|} \left[ \sum_{l=0}^{\infty} \lambda^l \mathbf{A}_\times^l \right]_{ij} = \sum_{i=1}^{|V_\times|} \sum_{i=j}^{|V_\times|} \left[ (\mathbf{I} - \lambda \mathbf{A}_\times)^{-1} \right]_{ij} = \mathbf{1}^\top (\mathbf{I} - \lambda \mathbf{A}_\times)^{-1} \mathbf{1} \qquad (2)$$

where $\mathbf{A}_\times = \mathbf{A} \otimes \mathbf{A}_i$ and $\mathbf{A}_i$ is the adjacency matrix of "hidden graph" $G_i$ obtained as $\mathbf{A}_i = f(\mathbf{W})$. Here, $f(\cdot)$ is a function whose output is non-negative and potentially bounded, i.e., $f(\mathbf{W}_i) = \mathrm{ReLU}(\mathbf{W}_i)$ or $f(\mathbf{W}_i) = \sigma(\mathbf{W}_i)$ where $\sigma(\cdot)$ denotes the sigmoid activation function. Then, given the set $\mathcal{G}_h = \{G_1, G_2, \ldots, G_N\}$ where $G_1, G_2, \ldots, G_N$ denote the $N$ "hidden graphs", we can compute $N$ kernel values in total. These kernel values can be thought of as features of the input graph, and can be concatenated to form a vector representation of the input graph. This vector can then be fed into a fully-connected neural network to produce the output.

Following Vishwanathan et al. (2010), to compute the geometric random walk graph kernel shown in Equation equation 2 above, we employ a two-step approach . We first need to solve the following linear system for $\mathbf{z}$:

$$(\mathbf{I} - \lambda \mathbf{A}_\times) \mathbf{z} = \mathbf{1}$$

Then, given $\mathbf{z}$, we can compute the kernel value as $k^\infty(G, G_i) = \mathbf{1}^\top \mathbf{z}$. To solve the above linear system, we capitalize on fixed point methods. We first rewrite the above system as:

$$\mathbf{z} = \mathbf{1} + \lambda \mathbf{A}_\times \mathbf{z} \qquad (3)$$

Now, solving for $\mathbf{z}$ is equivalent to finding a fixed point of Equation equation 3 (Nocedal & Wright, 2006). Such a fixed point can be obtained by simply iterating the first part of the forward pass. Letting $\mathbf{z}^{(t)}$ denote the value of $\mathbf{z}$ at iteration $t$, we set $\mathbf{z}^{(0)} = \mathbf{1}$, and then compute the following:

$$\mathbf{z}^{(t+1)} = \mathbf{1} + \lambda \mathbf{A}_\times \mathbf{z}^{(t)}$$

repeatedly until $||\mathbf{z}^{(t+1)} - \mathbf{z}^{(t)}|| < \epsilon$, where $||\cdot||$ denotes the Euclidean norm and $\epsilon$ some predefined tolerance or until a specific number of iterations has been reached. As mentioned in the previous section, the above problem is guaranteed to converge if the maximum eigenvalue of $\mathbf{A}_\times$ is strictly smaller than $1/\lambda$, thus if all the eigenvalues of $\lambda \mathbf{A}_\times$ lie inside the unit disk. If the values of the elements of $\mathbf{A}_i$ are bounded, we can compute an upper bound on the maximum degree of $G_\times$ and set the parameter $\lambda$ to some value smaller than the inverse of the upper bound.

**Efficient implementation.**   If the input graph $G$ consists of $n$ vertices and a "hidden graph" $G_i$ consists of $c$ vertices, then $\mathbf{A}_\times$ is an $nc \times nc$ matrix. Thus, multiplying $\mathbf{A}_\times$ by some vector inside the fixed-point algorithm requires $\mathcal{O}(n^2 c^2)$ operations in total. Fortunately, to compute the kernel, it is not necessary to explicitly compute matrix $\mathbf{A}_\times$. Specifically, the Kronecker product and vec operator are linked by the well-known property (Bernstein, 2009):

$$\mathrm{vec}(\mathbf{A}\,\mathbf{B}\,\mathbf{C}) = (\mathbf{C}^\top \otimes \mathbf{A})\mathrm{vec}(\mathbf{B}) \qquad (4)$$

Then, let $\mathbf{Z} \in \mathbb{R}^{n \times c}$ be a matrix such that $\mathbf{Z} = \mathrm{vec}^{-1}(\mathbf{z})$. Recall also that $\mathbf{A}_\times = \mathbf{A} \otimes \mathbf{A}_i$. Based on the above and on Equation equation 4, we can write:

$$\mathbf{A}_\times \mathbf{z} = (\mathbf{A} \otimes \mathbf{A}_i)\mathrm{vec}(\mathbf{Z}) = \mathrm{vec}(\mathbf{A}_i \mathbf{Z} \mathbf{A}^\top) = \mathrm{vec}(\mathbf{A}_i \mathrm{vec}^{-1}(\mathbf{z})\mathbf{A}^\top) \qquad (5)$$

The above matrix-vector product can be computed in $\mathcal{O}(n^2 c)$ time in case $n > c$. If $\mathbf{A}$ is sparse, then it can be computed yet more efficiently. Furthermore, we do not need to compute and store matrix $\mathbf{A}_\times$ which might not be feasible due to high memory requirements. Then, instead of solving the system of Equation equation 3, we solve the following equivalent system:

$$\mathbf{z} = \mathbf{1} + \lambda \, \text{vec}(\mathbf{A}_i \, \text{vec}^{-1}(\mathbf{z}) \mathbf{A}^\top) \tag{6}$$

**Node attributes.** In many real-world problems, vertices of the input graphs are annotated with real-valued multi-dimensional vertex attributes. We next generalize the proposed model to graphs that contain such vertex attributes. Let $\mathbf{X} \in \mathbb{R}^{n \times d}$ denote the matrix that contains the vertex attributes of the input graph $G$. As already mentioned, we also associate a trainable matrix $\mathbf{Q}_i \in \mathbb{R}^{c \times d}$ to each "hidden graph" $G_i$, where $c$ is the number of vertices of $G_i$. Then, let $\mathbf{S} = \sigma(\mathbf{X}\mathbf{Q}_i^\top) \in \mathbb{R}^{c \times n}$ where $\sigma(\cdot)$ denotes the sigmoid function. The $(j,k)^{th}$ element of matrix $\mathbf{S}$ is equal to the inner product (followed by a sigmoid) between the attributes of the $j^{th}$ vertex of the input graph $G$ and the $k^{th}$ vertex of the "hidden graph" $G_i$. Roughly speaking, this matrix encodes the similarity between the attributes of the vertices of the two graphs. Note that instead of directly using matrix $\mathbf{X}$, we can first transform it into matrix $\tilde{\mathbf{X}}$ using a single- or a multi-layer perceptron. Let $\mathbf{s} = \text{vec}(\mathbf{S})$ where $\mathbf{s} \in \mathbb{R}^{nc}$. Each element of $\mathbf{s}$ corresponds to a vertex of $G_\times$ and quantifies the similarity between the attributes of the pair of vertices (i.e., one from $G$ and one from $G_i$) it represents. Then, we can compute the geometric random walk kernel as follows:

$$\begin{aligned}
k^\infty(G, G') &= \sum_{i=1}^{|V_\times|} \sum_{i=j}^{|V_\times|} \left[ \sum_{l=0}^{\infty} \lambda^l \big((\mathbf{s}\mathbf{s}^\top) \odot \mathbf{A}_\times\big)^l \right]_{ij} \\
&= \sum_{i=1}^{|V_\times|} \sum_{i=j}^{|V_\times|} \left[ \big(\mathbf{I} - \lambda(\mathbf{s}\mathbf{s}^\top) \odot \mathbf{A}_\times\big)^{-1} \right]_{ij} = \mathbf{1}^\top \big(\mathbf{I} - \lambda(\mathbf{s}\mathbf{s}^\top) \odot \mathbf{A}_\times\big)^{-1} \mathbf{1}
\end{aligned} \tag{7}$$

Note that since the elements of $\mathbf{s}$ take values between 0 and 1, the same applies to the elements of the output of the outer product $\mathbf{s}\mathbf{s}^\top$. Therefore, the maximum degree of the vertices of the graph derived from the matrix $\mathbf{s}\mathbf{s}^\top \odot \mathbf{A}_\times$ is not greater than that of the graph derived from matrix $\mathbf{A}_\times$, and we do not thus need to set $\lambda$ to a new value. Then, to compute the kernel, we first need to solve the following system:

$$\mathbf{z} = \mathbf{1} + \lambda(\mathbf{s}\mathbf{s}^\top \odot \mathbf{A}_\times)\mathbf{z} \tag{8}$$

Again, naively computing the right part of the above Equation is expensive and requires $\mathcal{O}(n^2 c^2)$ operations in total. The following result shows that in fact we can compute the above in a more time and memory efficient manner.

**Proposition 1.** *Let $\mathbf{A}_1 \in \mathbb{R}^{n \times n}$ and $\mathbf{A}_2 \in \mathbb{R}^{m \times m}$ be two real matrices. Let also $\mathbf{s}, \mathbf{y} \in \mathbb{R}^{nm}$ be two real vectors. Then, we have that:*

$$\big(\mathbf{s}\mathbf{s}^\top \odot (\mathbf{A}_1 \otimes \mathbf{A}_2)\big)\mathbf{y} = \mathbf{s} \odot vec\big(\mathbf{A}_2 \, vec^{-1}(\mathbf{y} \odot \mathbf{s})\mathbf{A}_1^\top\big)$$

Based on the above result (the proof is left to the appendix), the system that needs to be solved is:

$$\mathbf{z} = \mathbf{1} + \lambda\Big(\mathbf{s} \odot \text{vec}\big(\mathbf{A}_i \, \text{vec}^{-1}(\mathbf{z} \odot \mathbf{s})\mathbf{A}^\top\big)\Big)$$

Since we store matrix $\mathbf{A}$ as a sparse matrix, if there are $\mathcal{O}(m)$ non-zero entries in $\mathbf{A}$, then computing one iteration of the above equation for all $N$ "hidden graphs" takes $\mathcal{O}\big(Nc(n(d+c) + m)\big)$ computational time where $d$ is the size of the vertex attributes.

**Implicit differentiation.** Clearly, iteratively computing Equation equation 3 or Equation equation 8 to find the fixed point corresponds to a differentiable module. However, to train the model, we need to backpropagate the error through the fixed point solver in the backward pass. That would require storing all the intermediate terms, which could be prohibitive in practice. Fortunately, thanks to recent advances

in implicit layers and equilibrium models (Bai et al., 2019), this can be performed in a simple and efficient manner which requires constant memory, and assumes no knowledge of the fixed point solver. Specifically, based on ideas from Bai et al. (2019), we derive the form of implicit backpropagation specific to the employed fixed point iteration layer.

**Theorem 1.** *Let $f_\theta$ be the system of Equation equation 3 or Equation equation 8, and $\mathbf{z}^\star \in \mathbb{R}^{nc}$ be a solution to that linear system. Let also $g_\theta(\mathbf{z}^\star; \mathbf{A}, \mathbf{X}) = f_\theta(\mathbf{z}^\star; \mathbf{A}, \mathbf{X}) - \mathbf{z}^\star$. Since $\mathbf{z}^\star$ is a fixed point, we have that $g_\theta(\mathbf{z}^\star; \mathbf{A}, \mathbf{X}) \to 0$ and $\mathbf{z}^\star$ is thus the root of $g_\theta$. Let $y \in \mathbb{R}$ denote the ground-truth target of the input sample, $h : \mathbb{R} \to \mathbb{R}$ be any differentiable function and let $\mathcal{L} : \mathbb{R} \times \mathbb{R} \to \mathbb{R}$ be a loss function that computes:*

$$\ell = \mathcal{L}\big(h(\mathbf{1}^\top \mathbf{z}^\star), y\big) = \mathcal{L}\big(h\big(\mathbf{1}^\top \mathsf{FindRoot}(g_\theta; \mathbf{A}, \mathbf{X})\big), y\big) \tag{9}$$

*Then, the gradient of the loss w.r.t. $(\cdot)$ (e.g., $\theta$, $\mathbf{A}$ or $\mathbf{X}$) is:*

$$\frac{\partial \ell}{\partial(\cdot)} = -\frac{\partial \ell}{\partial \mathbf{z}^\star}\big(J_{g_\theta}^{-1}\big|_{\mathbf{z}^\star}\big)\frac{\partial f_\theta(\mathbf{z}^\star; \mathbf{A}, \mathbf{X})}{\partial(\cdot)} = -\frac{\partial \ell}{\partial h}\frac{\partial h}{\partial \mathbf{z}^\star}\big(J_{g_\theta}^{-1}\big|_{\mathbf{z}^\star}\big)\frac{\partial f_\theta(\mathbf{z}^\star; \mathbf{A}, \mathbf{X})}{\partial(\cdot)} \tag{10}$$

*where $J_{g_\theta}^{-1}\big|_{\mathbf{z}^\star}$ is the inverse Jacobian of $g_\theta$ evaluated at $\mathbf{z}^\star$.*

The above formula gives a form for the necessary Jacobian without needing to backpropagate through the method used to obtain the fixed point. Thus, as mentioned above, we only need to find the fixed point, and we can compute the necessary Jacobians at this specific point using the above analytical form. No intermediate terms of the iterative method used to compute the fixed point need to be stored in memory, while there is also no need to unroll the forward computations within an automatic differentiation layer. Still, to compute the analytical backward gradient at the solution of the fixed point equation, it is necessary to first compute the exact inverse Jacobian $J_{g_\theta}^{-1}$ which has a cubic cost. As shown in Bai et al. (2019), we can instead compute the $-\frac{\partial \ell}{\partial \mathbf{z}^\star}\big(J_{g_\theta}^{-1}\big|_{\mathbf{z}^\star}\big)$ term by solving the following linear system:

$$\mathbf{x} = \left(\frac{\partial f_\theta(\mathbf{z}^\star; \mathbf{A}, \mathbf{X})}{\partial \mathbf{z}^\star}\right)^\top \mathbf{x} + \left(\frac{\partial \ell}{\partial \mathbf{z}^\star}\right)^\top$$

which in fact is also a fixed point equation and can be solved via some iterative procedure. Note that the first term of the above Equation is a vector-Jacobian product which can be efficiently computed via autograd packages (e.g., PyTorch (Paszke et al., 2017)) for any $\mathbf{x}$, without explicitly writing out the Jacobian matrix. Finally, we can compute $\frac{\partial \ell}{\partial(\cdot)}$ as follows:

$$\frac{\partial \ell}{\partial(\cdot)} = \left(\frac{\partial f_\theta(\mathbf{z}^\star; \mathbf{A}, \mathbf{X})}{\partial(\cdot)}\right)^\top \mathbf{x}$$

where again this product is itself a vector-Jacobian product, computable via normal automatic differentiation packages.

## 5 Experimental Evaluation

We next evaluate the proposed GRWNN model on standard graph classification datasets.

### 5.1 Real-World Datasets

**Datasets.** We evaluate the proposed model on 10 publicly available graph classification datasets including 5 bio/chemo-informatics datasets: MUTAG, D&D, NCI1, PROTEINS, ENZYMES, and 5 social interaction datasets: IMDB-BINARY, IMDB-MULTI, REDDIT-BINARY, REDDIT-MULTI-5K, COLLAB (Kersting et al., 2016). To show that the proposed model also scales to larger datasets, we additionally use two Open Graph Benchmark (OGB) datasets (Hu et al., 2020). Specifically, we use a molecular property prediction dataset: `ogbg-molhiv`, and a code summarization dataset: `ogbg-code2`. More details about the datasets are given in the appendix.

**Experimental Setup.** In the case of the 10 standard benchmark datasets, we compare the proposed model against the following three graph kernels: (1) graphlet kernel (GR) (Shervashidze et al., 2009), (2) shortest path kernel (SP) (Borgwardt & Kriegel, 2005), and (3) Weisfeiler-Lehman subtree kernel (WL) (Shervashidze et al., 2011), and against the following six neural network models: (1) DGCNN (Zhang et al., 2018a), (2) DiffPool (Ying et al., 2018), (3) ECC (Simonovsky & Komodakis, 2017), (4) GIN (Xu et al., 2019), (5) GraphSAGE (Hamilton et al., 2017), and (6) RWNN (Nikolentzos & Vazirgiannis, 2020). We also compare the proposed model against GRWNN-fixed, a variant of the model whose "hidden graphs" are randomly initialized and kept fixed during training. To evaluate the proposed model, we employ the experimental protocol proposed in (Errica et al., 2020). Therefore, we perform 10-fold cross-validation to obtain an estimate of the generalization performance of each method, while within each fold a model is selected based on a 90%/10% split of the training set. We use exactly the same splits as in (Errica et al., 2020) and in (Nikolentzos & Vazirgiannis, 2020), hence, for the different datasets, we use the results reported in these two papers.

For all datasets, we set the batch size to 64 and the number of epochs to 300. We use the Adam optimizer with initial learning rate 0.001 and applied an adaptive learning rate decay based on validation results. We use a 1-layer perceptron to transform the vertex attributes. We apply layer normalization (Ba et al., 2016) on the generated graph representations (i.e., vector consisting of kernel values). The hyper-parameters we tune for each dataset are: (1) the number of "hidden graphs" $\in \{32, 64\}$, (2) the number of vertices of the "hidden graphs" $\in \{5, 10\}$, (3) the hidden-dimension size of the vertex features $\in \{32, 64\}$ for the bio/chemo-informatics datasets and $\in \{8, 16\}$ for the social interaction datasets, and (4) the dropout ratio $\in \{0, 0.1\}$.

For both OGB datasets, we used the available predefined splits. We compare the proposed model against the following neural network models: GCN (Kipf & Welling, 2017), GIN (Xu et al., 2019), GCN-FLAG (Kong et al., 2020), GIN-FLAG (Kong et al., 2020), PNA (Corso et al., 2020), GSN (Bouritsas et al., 2020), HIMP (Fey et al., 2020), and DGN (Beaini et al., 2020). For all models, we use the results that are reported in the respective papers. For `ogbg-code2`, we did not add the inverse edges to the graphs. All reported results are averaged over 10 runs.

For both OGB datasets, we set the batch size to 128. For the `ogb-molhiv` dataset, we set the number of epochs to 300, the number of "hidden graphs" to 200, the number of vertices of the "hidden graphs" to 5, the hidden-dimension size of the vertex features to 128 and the dropout ratio to 0.1. Furthermore, we employ the probabilistic variant of the geometric random walk kernel and use uniform distributions for the initial and stopping probabilities over the vertices of the two compared graphs. For the `ogb-code2` dataset, we set the number of epochs to 100, the number of "hidden graphs" to 200, the number of vertices of the "hidden graphs" to 5, the hidden-dimension size of the vertex features to 128 and the dropout ratio to 0.2. For both datasets, we apply layer normalization (Ba et al., 2016) on the generated graph representations.

**Implementation Details.** To set the value of parameter $\lambda$, we assume a transductive setting, where we are given a collection of graphs beforehand. Therefore, we can find the vertex of highest degree across all graphs and set the value of $\lambda$ accordingly. In the inductive learning setting, since we do not know a priori target graphs that the model may receive in the future, $\lambda$ should be small enough so that $\lambda < 1/\mu_\times^{max}$ for any pair of an unseen graph and a "hidden graph". This is a limitation of the proposed model since in case the model receives at test time a graph whose largest eigenvalue is higher than expected, we need to set $\lambda$ to a smaller value and retrain the model.

To compute the fixed point of Equation equation 3 or equation 8, we followed the naive approach where we simply performed multiple times the forward iteration. In practice, there are more efficient fixed point iteration methods, such as Anderson Acceleration (Walker & Ni, 2011), that converge faster than the naive forward iteration at the cost of some additional memory complexity. However, as shown next, we found that in our setting, the naive forward iteration converges in a small number of steps, while the additional cost introduced by more efficient methods associated with the generation and manipulation of new tensors made them overall slower than the naive forward iteration even though they required fewer iterations to converge.

The model was implemented with PyTorch (Paszke et al., 2019), and all experiments were run on a single machine equipped with an NVidia Titan Xp GPU card.

Table 1: Classification accuracy ($\pm$ standard deviation) of the proposed model and the baselines on the 5 chemo/bio-informatics and on the 5 social interaction benchmark datasets. OOR means Out of Resources, either time ($>$ 72 hours for a single training) or GPU memory. Best performance per dataset in **bold**, among the neural network architectures underlined.

| | MUTAG | D&D | NCI1 | PROTEINS | ENZYMES |
|---|---|---|---|---|---|
| SP | 80.2 ($\pm$ 6.5) | **78.1** ($\pm$ 4.1) | 72.7 ($\pm$ 1.4) | **75.3** ($\pm$ 3.8) | 38.3 ($\pm$ 8.0) |
| GR | 80.8 ($\pm$ 6.4) | 75.4 ($\pm$ 3.4) | 61.8 ($\pm$ 1.7) | 71.6 ($\pm$ 3.1) | 25.1 ($\pm$ 4.4) |
| WL | 84.6 ($\pm$ 8.3) | **78.1** ($\pm$ 2.4) | **84.8** ($\pm$ 2.5) | 73.8 ($\pm$ 4.4) | 50.3 ($\pm$ 5.7) |
| DGCNN | 84.0 ($\pm$ 6.7) | 76.6 ($\pm$ 4.3) | 76.4 ($\pm$ 1.7) | 72.9 ($\pm$ 3.5) | 38.9 ($\pm$ 5.7) |
| DiffPool | 79.8 ($\pm$ 7.1) | 75.0 ($\pm$ 3.5) | 76.9 ($\pm$ 1.9) | 73.7 ($\pm$ 3.5) | 59.5 ($\pm$ 5.6) |
| ECC | 75.4 ($\pm$ 6.2) | 72.6 ($\pm$ 4.1) | 76.2 ($\pm$ 1.4) | 72.3 ($\pm$ 3.4) | 29.5 ($\pm$ 8.2) |
| GIN | 84.7 ($\pm$ 6.7) | 75.3 ($\pm$ 2.9) | 80.0 ($\pm$ 1.4) | 73.3 ($\pm$ 4.0) | 59.6 ($\pm$ 4.5) |
| GraphSAGE | 83.6 ($\pm$ 9.6) | 72.9 ($\pm$ 2.0) | 76.0 ($\pm$ 1.8) | 73.0 ($\pm$ 4.5) | 58.2 ($\pm$ 6.0) |
| 1-step RWNN | **89.2** ($\pm$ 4.3) | 77.6 ($\pm$ 4.7) | 71.4 ($\pm$ 1.8) | 74.7 ($\pm$ 3.3) | 56.7 ($\pm$ 5.2) |
| 2-step RWNN | 88.1 ($\pm$ 4.8) | 76.9 ($\pm$ 4.6) | 73.0 ($\pm$ 2.0) | 74.1 ($\pm$ 2.8) | 57.4 ($\pm$ 4.9) |
| GRWNN-fixed | 81.9 ($\pm$ 6.4) | 73.2 ($\pm$ 3.5) | 66.9 ($\pm$ 2.4) | 74.6 ($\pm$ 4.0) | 56.8 ($\pm$ 3.7) |
| GRWNN | 83.4 ($\pm$ 5.6) | 75.6 ($\pm$ 4.6) | 67.7 ($\pm$ 2.2) | 74.9 ($\pm$ 3.5) | **62.7** ($\pm$ 5.2) |

| | IMDB BINARY | IMDB MULTI | REDDIT BINARY | REDDIT MULTI-5K | COLLAB |
|---|---|---|---|---|---|
| SP | 57.7 ($\pm$ 4.1) | 39.8 ($\pm$ 3.7) | 89.0 ($\pm$ 1.0) | 51.1 ($\pm$ 2.2) | **79.9** ($\pm$ 2.7) |
| GR | 63.3 ($\pm$ 2.7) | 39.6 ($\pm$ 3.0) | 76.6 ($\pm$ 3.3) | 38.1 ($\pm$ 2.3) | 71.1 ($\pm$ 1.4) |
| WL | **72.8** ($\pm$ 4.5) | **51.2** ($\pm$ 6.5) | 74.9 ($\pm$ 1.8) | 49.6 ($\pm$ 2.0) | 78.0 ($\pm$ 2.0) |
| DGCNN | 69.2 ($\pm$ 3.0) | 45.6 ($\pm$ 3.4) | 87.8 ($\pm$ 2.5) | 49.2 ($\pm$ 1.2) | 71.2 ($\pm$ 1.9) |
| DiffPool | 68.4 ($\pm$ 3.3) | 45.6 ($\pm$ 3.4) | 89.1 ($\pm$ 1.6) | 53.8 ($\pm$ 1.4) | 68.9 ($\pm$ 2.0) |
| ECC | 67.7 ($\pm$ 2.8) | 43.5 ($\pm$ 3.1) | OOR | OOR | OOR |
| GIN | 71.2 ($\pm$ 3.9) | 48.5 ($\pm$ 3.3) | 89.9 ($\pm$ 1.9) | **56.1** ($\pm$ 1.7) | 75.6 ($\pm$ 2.3) |
| GraphSAGE | 68.8 ($\pm$ 4.5) | 47.6 ($\pm$ 3.5) | 84.3 ($\pm$ 1.9) | 50.0 ($\pm$ 1.3) | 73.9 ($\pm$ 1.7) |
| 1-step RWNN | 70.8 ($\pm$ 4.8) | 47.8 ($\pm$ 3.8) | **90.4** ($\pm$ 1.9) | 51.7 ($\pm$ 1.5) | 71.7 ($\pm$ 2.1) |
| 2-step RWNN | 70.6 ($\pm$ 4.4) | 48.8 ($\pm$ 2.9) | 90.3 ($\pm$ 1.8) | 51.7 ($\pm$ 1.4) | 71.3 ($\pm$ 2.1) |
| GRWNN-fixed | 72.1 ($\pm$ 4.1) | 48.1 ($\pm$ 3.6) | 82.2 ($\pm$ 2.4) | 53.1 ($\pm$ 1.8) | 71.3 ($\pm$ 1.9) |
| GRWNN | **72.8** ($\pm$ 4.2) | 49.0 ($\pm$ 2.9) | 90.0 ($\pm$ 1.8) | 54.4 ($\pm$ 1.7) | 72.1 ($\pm$ 1.9) |

**Results.** Table 1 illustrates average prediction accuracies and standard deviations for the 10 standard graph classification datasets. We observe that the proposed GRWNN model is the best-performing method on 2 out of the 10 datasets, while it provides the second best and third best accuracy on 3 and 1 out of the remaining 8 datasets, respectively. The most successful method is the WL kernel which performs best on 4 of the 10 datasets, while it outperforms the other approaches with quite wide margins in most cases. Among the neural network models, the proposed GRWNN model outperforms the baseline models on 4 out of the 10 datasets. On the remaining 6 datasets, GIN is the best-performing model on half of them, and RWNN on the other half. On the ENZYMES and IMDB-BINARY datasets, our model offers respective absolute improvements of 3.1%, and 1.6% in accuracy over GIN. Overall, the model exhibits highly competitive performance on the graph classification datasets, while the achieved accuracies follow different patterns from all the baseline methods. Furthermore, the proposed model outperforms GRWNN-fixed on all datasets, demonstrating that the set of trainable "hidden graphs" is an indispensable component of the model.

The Table shown in Figure 1a illustrates the performance on the two OGB datasets. Note that the proposed model does not utilize the edge features that are provided for the different datasets. Still, we can see that it managed to outperform several of the baselines on the `ogbg-molhiv` dataset, where it achieved the fourth best ROC-AUC. On the `ogbg-code2` dataset, GRWNN outperformed GIN, while it achieved an F1-score similar to that of GCN. However, all these three models achieved a much smaller F1-score than the one achieved by PNA which is the best-performing model.

As already discussed, the running time of the model depends on the number of fixed point iterations that need to be performed until convergence. Figure 1b (top) illustrates the average number of iterations (across all batches) for the forward and backward pass for different values of $\lambda$ and for each epoch. The model was trained on a single split of the ENZYMES dataset. The maximum eigenvalue of all graphs of the dataset is

(a) Performance of the proposed model and the baselines on the OGB datasets. Reported values correspond to ROC-AUC scores for `ogbg-molhiv` and F1-scores for `ogbg-code2`.

| Method | Dataset | |
|---|---|---|
| | ogbg-molhiv | ogbg-code2 |
| GCN | $76.06 \pm 0.97$ | $15.07 \pm 0.18$ |
| GIN | $75.58 \pm 1.40$ | $14.95 \pm 0.23$ |
| GCN+ FLAG | $76.83 \pm 1.02$ | – |
| GIN+ FLAG | $76.54 \pm 1.14$ | – |
| GSN | $77.99 \pm 1.00$ | – |
| HIMP | $78.80 \pm 0.82$ | – |
| PNA | $79.05 \pm 1.32$ | $15.70 \pm 0.32$ |
| DGN | $79.70 \pm 0.97$ | – |
| GRWNN | $78.38 \pm 0.99$ | $15.03 \pm 0.21$ |

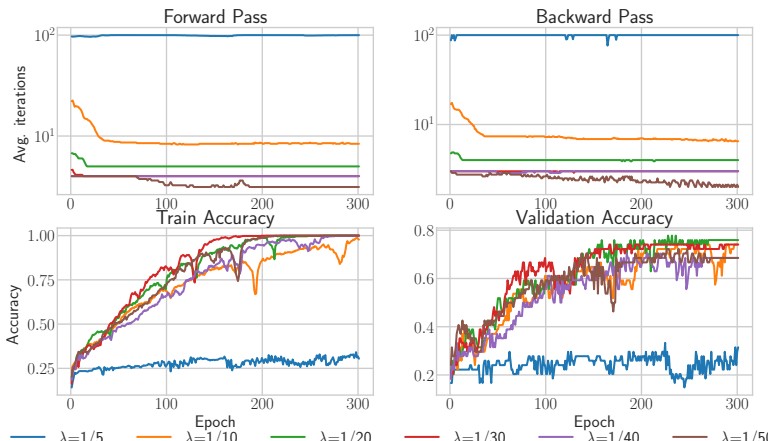

(b) Number of fixed point iterations during the forward and backward pass (top), and training and validation accuracy (bottom) on the EN-ZYMES dataset for different values of $\lambda$.

Figure 1: Performance on the OGB datasets and impact of the value of parameter $\lambda$ on running time and performance of the model.

Table 2: Average running time per epoch (in seconds) of the proposed model and 3 baselines on the 10 graph classification datasets.

| | MUTAG | D&D | NCI1 | PROTEINS | ENZYMES | IMDB BINARY | IMDB MULTI | REDDIT BINARY | REDDIT MULTI-5K | COLLAB |
|---|---|---|---|---|---|---|---|---|---|---|
| GIN | 0.03 | 0.34 | 0.50 | 0.14 | 0.07 | 0.13 | 0.19 | 0.81 | 2.43 | 0.98 |
| 2-RWNN | 0.03 | 0.19 | 0.57 | 0.16 | 0.08 | 0.14 | 0.20 | 0.43 | 1.13 | 0.89 |
| 3-RWNN | 0.04 | 0.23 | 0.76 | 0.21 | 0.11 | 0.18 | 0.28 | 0.55 | 1.42 | 1.04 |
| GRWNN | 0.07 | 0.77 | 0.94 | 0.32 | 0.17 | 0.24 | 0.34 | 2.93 | 6.19 | 2.69 |

equal to 5.47, while the highest degree is equal to 9. The number of nodes of the "hidden graphs" was set to 5. If the elements of the adjacency matrices of the "hidden graphs" take values no greater than one, then no vertex of $G_\times$ can have a degree greater than $9 * 4 = 36$. Thus, setting $\lambda < 1/36$ guarantees convergence. In practice, as shown in the Figure, we found that even if $\lambda$ takes larger values, we only need a small number of iterations. For $\lambda = 1/5$, we can see that the fixed point equation fails to converge since the average number of iterations is close to 100 (which is the upper limit we have set). For $\lambda = 1/10$ and for smaller values of $\lambda$, the system converges in a small number of iterations. In terms of performance, as shown in Figure 1b (bottom), the model achieves the highest levels of validation accuracy for $\lambda = 1/20$ and $\lambda = 1/30$, while for $\lambda = 1/5$, the model yields much worse performance compared to the other values of $\lambda$. Similar behavior was observed on the other datasets.

## 5.2 Runtime Analysis

The proposed model is indeed computationally more expensive than the RWNN model due to the fixed point iteration which is not parallelizable. However, as already discussed, we empirically observed that the forward iteration converges in a small number of steps, thus incurring a relatively small overhead in the model's running time. We have computed the average running time per epoch of the proposed model, and 3 of the baselines (2-RWNN, 3-RWNN and GIN) on the 10 graph classification datasets. We use the same values for the common hyperparameters (e.g., number and size of hidden graphs for GRWNN and RWNN, and hidden dimension size, batch size, etc for all 3 models). The results are shown in Table 2 (in seconds). As we can see, the proposed model is not much more expensive than the baselines. In fact, in most cases,

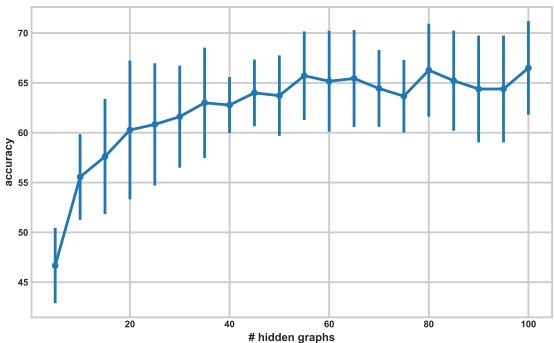 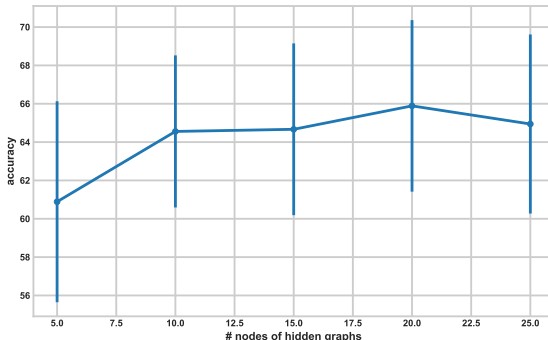

Figure 2: Performance on the ENZYMES dataset as a function of the number of "hidden graphs" (left) and the number of vertices of the "hidden graphs" (right).

its average running time per epoch is $1 - 3$ times higher than that of the baselines, which is by no means prohibitive for real-world scenarios.

### 5.3 Sensitivity Analysis

The proposed GRWNN model involves two main parameters: (1) the number of "hidden graphs", and (2) the number of vertices of "hidden graphs". We next investigate how these two parameters influence the performance of the GRWNN model. Specifically, in Figure 2, we examine how the different values of these parameters affect the performance of GRWNN on the ENZYMES dataset. We observe that the accuracy on the test set increases as the number of "hidden graphs" increases. The number of "hidden graphs" seems to have a significant impact on the performance of the model. When the number of graphs is set equal to 5, the model achieves an accuracy smaller than 50%, while when the number of graphs is set equal to 100, it yields an accuracy greater than 65%. On the other hand, the number of vertices of the "hidden graphs" does not affect that much the performance of the model.

## 6 Conclusion

In this paper, we introduced the GRWNN model, a new architecture which generates graph representations by comparing the input graphs against "hidden graphs" using the geometric random walk kernel. To compute the kernel, the proposed model uses a fixed point iteration algorithm, and to update the "hidden graphs" during backpropagation, the model capitalizes on implicit differentation and employs an efficient training scheme which requires only constant memory, regardless of the number of fixed-point iterations. The model was evaluated on several graph classification datasets where it achieved competitive performance.

The main cotribution of this work is methodological and therefore, there are no negative societal impacts directly related to it. Although we are not aware of any malicious uses of GNNs so far, these models could potentially serve as the key component behind harmful applications. For instance, in a social network where vertices represent humans, one could use GNNs to discriminate people in terms of some desired characteristic which can potentially affect people and their rights. Still, we believe that the benefits that arise from the use of these models (e. g., drug discovery) outweigh the potential negative societal impacts. To mitigate the risks associated with the use of GNNs, the community could develop tools that can identify potential harms. For instance, the negative impact of the aforementioned harmful application could be mitigated by developing tools that can detect whether individuals or groups of people are subject to discrimination.

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

## A   Appendix

The appendix is organized as follows. In section B, we define some basic concepts from linear algebra. In section C, we prove the Proposition 1. In section D, we give more details about the direct differentiation through the fixed point, while section E provides a detailed description of the graph classification datasets. Finally, section F gives details about the parameter $\lambda$.

## B   Linear Algebra Concepts

In this Section, we provide definitions for concepts of linear algebra, namely the vectorization operator, the inverse vectorization operator and the Kronecker product, which we use heavily in the main paper.

**Definition 1.** *Given a real matrix $\mathbf{A} \in \mathbb{R}^{m \times n}$, the vectorization operator $vec : \mathbb{R}^{m \times n} \to \mathbb{R}^{mn}$ is defined as:*

$$
vec(\mathbf{A}) = \begin{bmatrix} \mathbf{A}_{:1} \\ \mathbf{A}_{:2} \\ \vdots \\ \mathbf{A}_{:n} \end{bmatrix}
$$

*where $\mathbf{A}_{:i}$ is the $i^{th}$ column of $\mathbf{A}$.*

**Definition 2.** *Given a real vector* $\mathbf{b} \in \mathbb{R}^{mn}$, *the inverse vectorization operator* $vec^{-1} : \mathbb{R}^{nm} \rightarrow \mathbb{R}^{n \times m}$ *is defined as:*

$$vec^{-1}(\mathbf{b}) = \begin{bmatrix} \mathbf{b}_1 & \mathbf{b}_{n+1} & \dots & \mathbf{b}_{n(m-1)+1} \\ \mathbf{b}_2 & \mathbf{b}_{n+2} & \dots & \mathbf{b}_{n(m-1)+2} \\ \vdots & \vdots & \vdots & \vdots \\ \mathbf{b}_n & \mathbf{b}_{2n} & \dots & \mathbf{b}_{nm} \end{bmatrix}$$

**Definition 3.** *Given real matrices* $\mathbf{A} \in \mathbb{R}^{n \times m}$ *and* $\mathbf{B} \in \mathbb{R}^{p \times q}$, *the Kronecker product* $\mathbf{A} \otimes \mathbf{B} \in \mathbb{R}^{np \times mq}$ *defined as:*

$$\mathbf{A} \otimes \mathbf{B} = \begin{bmatrix} \mathbf{A}_{11}\,\mathbf{B} & \mathbf{A}_{12}\,\mathbf{B} & \dots & \mathbf{A}_{1m}\,\mathbf{B} \\ \mathbf{A}_{21}\,\mathbf{B} & \mathbf{A}_{22}\,\mathbf{B} & \dots & \mathbf{A}_{2m}\,\mathbf{B} \\ \vdots & \vdots & \vdots & \vdots \\ \mathbf{A}_{n1}\,\mathbf{B} & \mathbf{A}_{n2}\,\mathbf{B} & \dots & \mathbf{A}_{nm}\,\mathbf{B} \end{bmatrix}$$

## C   Proof of Proposition 1

For convenience, we restate the Proposition below.

**Proposition 2.** *Let* $\mathbf{A}_1 \in \mathbb{R}^{n \times n}$ *and* $\mathbf{A}_2 \in \mathbb{R}^{m \times m}$ *be two real matrices. Let also* $\mathbf{s}, \mathbf{y} \in \mathbb{R}^{nm}$ *be two real vectors. Then, we have that:*

$$\left(\mathbf{s}\,\mathbf{s}^\top \odot (\mathbf{A}_1 \otimes \mathbf{A}_2)\right)\mathbf{y} = \mathbf{s} \odot vec\left(\mathbf{A}_2\, vec^{-1}(\mathbf{s} \odot \mathbf{y})\mathbf{A}_1^\top\right)$$

*Proof.* Let $\mathbf{D}_\mathbf{s}$ denote a diagonal matrix with the vector $\mathbf{s}$ as its main diagonal. Then, we have:

$$\left(\mathbf{s}\,\mathbf{s}^\top \odot (\mathbf{A}_1 \otimes \mathbf{A}_2)\right)\mathbf{y} = \left(\mathbf{D}_\mathbf{s}\,(\mathbf{A}_1 \otimes \mathbf{A}_2)\,\mathbf{D}_\mathbf{s}\right)\mathbf{y}$$

The Hadamard product of two vectors $\mathbf{s}$ and $\mathbf{y}$ is the same as matrix multiplication of vector $\mathbf{y}$ by the corresponding diagonal matrix $\mathbf{D}_\mathbf{s}$ of vector $\mathbf{s}$, i.e., $\mathbf{D}_\mathbf{s}\,\mathbf{y} = \mathbf{s} \odot \mathbf{y}$. Thus, it follows that:

$$\left(\mathbf{D}_\mathbf{s}\,(\mathbf{A}_1 \otimes \mathbf{A}_2)\,\mathbf{D}_\mathbf{s}\right)\mathbf{y} = \mathbf{D}_\mathbf{s}\,(\mathbf{A}_1 \otimes \mathbf{A}_2)\,(\mathbf{s} \odot \mathbf{y})$$

Note that the Kronecker product and vec operator are linked by the well-known property (Bernstein, 2009)(Proposition 7.1.9):

$$\text{vec}(\mathbf{A}\,\mathbf{B}\,\mathbf{C}) = (\mathbf{C}^\top \otimes \mathbf{A})\text{vec}(\mathbf{B})$$

Therefore, we have that:

$$\left(\mathbf{D}_\mathbf{s}\,(\mathbf{A}_1 \otimes \mathbf{A}_2)\,(\mathbf{s} \odot \mathbf{y}) = \mathbf{D}_\mathbf{s}\,\text{vec}\left(\mathbf{A}_2\,\text{vec}^{-1}(\mathbf{s} \odot \mathbf{y})\,\mathbf{A}_1^\top\right) = \mathbf{s} \odot \text{vec}\left(\mathbf{A}_2\,\text{vec}^{-1}(\mathbf{s} \odot \mathbf{y})\,\mathbf{A}_2^\top\right)\right.$$

which concludes the proof. $\qquad\qquad\square$

## D   Implicit Differentiation

Clearly, iteratively computing Equation (3) or Equation (8) (main paper) to find the fixed point corresponds to a differentiable module. However, to train the model, we need to backpropagate the error through the fixed point solver in the backward pass. That would require storing all the intermediate terms, which could be prohibitive in practice. Fortunately, thanks to recent advances in implicit layers and equilibrium models (Bai et al., 2019), this can be performed in a simple and efficient manner which requires constant memory, and assumes no knowledge of the fixed point solver. Specifically, based on ideas from (Bai et al., 2019), we derive the form of implicit backpropagation specific to the employed fixed point iteration layer.

**Theorem 2.** *Let $f_\theta$ be the system of Equation (3) or Equation (8) (main paper), and $\mathbf{z}^\star \in \mathbb{R}^{nc}$ be a solution to that linear system. Let also $g_\theta(\mathbf{z}^\star; \mathbf{A}, \mathbf{X}) = f_\theta(\mathbf{z}^\star; \mathbf{A}, \mathbf{X}) - \mathbf{z}^\star$. Since $\mathbf{z}^\star$ is a fixed point, we have that $g_\theta(\mathbf{z}^\star; \mathbf{A}, \mathbf{X}) \to 0$ and $\mathbf{z}^\star$ is thus the root of $g_\theta$. Let $y \in \mathbb{R}$ denote the ground-truth target of the input sample, $h : \mathbb{R} \to \mathbb{R}$ be any differentiable function and let $\mathcal{L} : \mathbb{R} \times \mathbb{R} \to \mathbb{R}$ be a loss function that computes:*

$$\ell = \mathcal{L}\big(h(\mathbf{1}^\top \mathbf{z}^\star), y\big) = \mathcal{L}\big(h\big(\mathbf{1}^\top \mathsf{FindRoot}(g_\theta; \mathbf{A}, \mathbf{X})\big), y\big) \tag{11}$$

*Then, the gradient of the loss w.r.t. $(\cdot)$ (e.g., $\theta$, $\mathbf{A}$ or $\mathbf{X}$) is:*

$$\frac{\partial \ell}{\partial (\cdot)} = -\frac{\partial \ell}{\partial \mathbf{z}^\star}\big(J_{g_\theta}^{-1}\big|_{\mathbf{z}^\star}\big)\frac{\partial f_\theta(\mathbf{z}^\star; \mathbf{A}, \mathbf{X})}{\partial (\cdot)} = -\frac{\partial \ell}{\partial h}\frac{\partial h}{\partial \mathbf{z}^\star}\big(J_{g_\theta}^{-1}\big|_{\mathbf{z}^\star}\big)\frac{\partial f_\theta(\mathbf{z}^\star; \mathbf{A}, \mathbf{X})}{\partial (\cdot)} \tag{12}$$

*where $J_{g_\theta}^{-1}\big|_{\mathbf{z}^\star}$ is the inverse Jacobian of $g_\theta$ evaluated at $\mathbf{z}^\star$.*

The above Theorem gives a form for the necessary Jacobian without needing to backpropagate through the method used to obtain the fixed point. We can thus treat the fixed point algorithm as a black box, and we do not need to store intermediate terms associated with the fixed point algorithm into memory. We only need to apply some algorithm that will produce a solution to the system (i. e., it will compute the fixed point).

Following (Bai et al., 2019), we differentiate the two sides of the fixed point equation $\mathbf{z}^\star = f_\theta(\mathbf{z}^\star; \mathbf{A}, \mathbf{X})$ wih respect to $(\cdot)$:

$$\frac{\mathrm{d}\mathbf{z}^\star}{\mathrm{d}(\cdot)} = \frac{\mathrm{d}f_\theta(\mathbf{z}^\star; \mathbf{A}, \mathbf{X})}{\mathrm{d}(\cdot)} = \frac{\partial f_\theta(\mathbf{z}^\star; \mathbf{A}, \mathbf{X})}{\partial (\cdot)} + \frac{\partial f_\theta(\mathbf{z}^\star; \mathbf{A}, \mathbf{X})}{\partial \mathbf{z}^\star}\frac{\mathrm{d}\mathbf{z}^\star}{\mathrm{d}(\cdot)}$$

$$\implies \left(\mathbf{I} - \frac{\partial f_\theta(\mathbf{z}^\star; \mathbf{A}, \mathbf{X})}{\partial \mathbf{z}^\star}\right)\frac{\mathrm{d}\mathbf{z}^\star}{\mathrm{d}(\cdot)} = \frac{\partial f_\theta(\mathbf{z}^\star; \mathbf{A}, \mathbf{X})}{\partial (\cdot)}$$

Since $g_\theta(\mathbf{z}^\star) = f_\theta(\mathbf{z}^\star; \mathbf{A}, \mathbf{X}) - \mathbf{z}^\star$, we have:

$$J_{g_\theta}\big|_{\mathbf{z}^\star} = -\left(\mathbf{I} - \frac{\partial f_\theta(\mathbf{z}^\star; \mathbf{A}, \mathbf{X})}{\partial \mathbf{z}^\star}\right)$$

which implies the following:

$$\frac{\partial \ell}{\partial (\cdot)} = \frac{\partial \ell}{\partial \mathbf{z}^\star}\frac{\mathrm{d}\mathbf{z}^\star}{\mathrm{d}(\cdot)} = -\frac{\partial \ell}{\partial \mathbf{z}^\star}\big(J_{g_\theta}^{-1}\big|_{\mathbf{z}^\star}\big)\frac{\partial f_\theta(\mathbf{z}^\star; \mathbf{A}, \mathbf{X})}{\partial (\cdot)}.$$

Unfortunately, computing the exact inverse Jacobian $J_{g_\theta}^{-1}$ has a cubic cost. As shown in (Bai et al., 2019), we can instead compute the $-\frac{\partial \ell}{\partial \mathbf{z}^\star}\big(J_{g_\theta}^{-1}\big|_{\mathbf{z}^\star}\big)$ term of the gradient (which contains the Jacobian) by solving the following linear system:

$$\mathbf{x}^\top = -\frac{\partial \ell}{\partial \mathbf{z}^\star}\big(J_{g_\theta}^{-1}\big|_{\mathbf{z}^\star}\big) = \frac{\partial \ell}{\partial \mathbf{z}^\star}\left(\mathbf{I} - \frac{\partial f_\theta(\mathbf{z}^\star; \mathbf{A}, \mathbf{X})}{\partial \mathbf{z}^\star}\right)^{-1}$$

$$\implies \mathbf{x} = \left(\left(\mathbf{I} - \frac{\partial f_\theta(\mathbf{z}^\star; \mathbf{A}, \mathbf{X})}{\partial \mathbf{z}^\star}\right)^\top\right)^{-1}\left(\frac{\partial \ell}{\partial \mathbf{z}^\star}\right)^\top$$

$$\implies \left(\mathbf{I} - \frac{\partial f_\theta(\mathbf{z}^\star; \mathbf{A}, \mathbf{X})}{\partial \mathbf{z}^\star}\right)^\top \mathbf{x} = \left(\frac{\partial \ell}{\partial \mathbf{z}^\star}\right)^\top$$

$$\implies \mathbf{x} = \left(\frac{\partial f_\theta(\mathbf{z}^\star; \mathbf{A}, \mathbf{X})}{\partial \mathbf{z}^\star}\right)^\top \mathbf{x} + \left(\frac{\partial \ell}{\partial \mathbf{z}^\star}\right)^\top$$

which in fact is also a fixed point equation and can be solved via some iterative procedure. Note that the first term of the above Equation is a vector-Jacobian product which can be efficiently computed via autograd packages (e. g., PyTorch (Paszke et al., 2017)) for any $\mathbf{x}$, without explicitly writing out the Jacobian matrix. Finally, we can compute $\frac{\partial \ell}{\partial (\cdot)}$ as follows:

$$\frac{\partial \ell}{\partial (\cdot)} = \left(\frac{\partial f_\theta(\mathbf{z}^\star; \mathbf{A}, \mathbf{X})}{\partial (\cdot)}\right)^\top \mathbf{x}$$

Table 3: Summary of the 10 datasets that were used in our experiments.

| Dataset | MUTAG | D&D | NCI1 | PROTEINS | ENZYMES | IMDB BINARY | IMDB MULTI | REDDIT BINARY | REDDIT MULTI-5K | COLLAB |
|---|---|---|---|---|---|---|---|---|---|---|
| Max # vertices | 28 | 5,748 | 111 | 620 | 126 | 136 | 89 | 3,782 | 3,648 | 492 |
| Min # vertices | 10 | 30 | 3 | 4 | 2 | 12 | 7 | 6 | 22 | 32 |
| Average # vertices | 17.93 | 284.32 | 29.87 | 39.05 | 32.63 | 19.77 | 13.00 | 429.61 | 508.50 | 74.49 |
| Max # edges | 33 | 14,267 | 119 | 1,049 | 149 | 1,249 | 1,467 | 4,071 | 4,783 | 40,119 |
| Min # edges | 10 | 63 | 2 | 5 | 1 | 26 | 12 | 4 | 21 | 60 |
| Average # edges | 19.79 | 715.66 | 32.30 | 72.81 | 62.14 | 96.53 | 65.93 | 497.75 | 594.87 | 2,457.34 |
| # labels | 7 | 82 | 37 | 3 | – | – | – | – | – | – |
| # attributes | – | – | – | – | 18 | – | – | – | – | – |
| # graphs | 188 | 1,178 | 4,110 | 1,113 | 600 | 1,000 | 1,500 | 2,000 | 4,999 | 5,000 |
| # classes | 2 | 2 | 2 | 2 | 6 | 2 | 3 | 2 | 5 | 3 |

where again this product is itself a vector-Jacobian product, computable via normal automatic differentiation packages.

# E  Datasets

We evaluated the proposed model on 10 publicly available graph classification datasets including 5 bio/chemo-informatics datasets: MUTAG, D&D, NCI1, PROTEINS and ENZYMES, as well as 5 social interaction datasets: IMDB-BINARY, IMDB-MULTI, REDDIT-BINARY, REDDIT-MULTI-5K and COLLAB (Kersting et al., 2016). A summary of the 10 datasets is given in Table 3. MUTAG consists of 188 mutagenic aromatic and heteroaromatic nitro compounds. The task is to predict whether or not each chemical compound has mutagenic effect on the Gram-negative bacterium *Salmonella typhimurium* (Debnath et al., 1991). ENZYMES contains 600 protein tertiary structures represented as graphs obtained from the BRENDA enzyme database. Each enzyme is a member of one of the Enzyme Commission top level enzyme classes (EC classes) and the task is to correctly assign the enzymes to their classes (Borgwardt et al., 2005). NCI1 contains more than four thousand chemical compounds screened for activity against non-small cell lung cancer and ovarian cancer cell lines (Wale et al., 2008). PROTEINS contains proteins represented as graphs where vertices are secondary structure elements and there is an edge between two vertices if they are neighbors in the amino-acid sequence or in 3D space. The task is to classify proteins into enzymes and non-enzymes (Borgwardt et al., 2005). D&D contains over a thousand protein structures. Each protein is a graph whose nodes correspond to amino acids and a pair of amino acids are connected by an edge if they are less than 6 Ångstroms apart. The task is to predict if a protein is an enzyme or not (Dobson & Doig, 2003). IMDB-BINARY and IMDB-MULTI were created from IMDb, an online database of information related to movies and television programs. The graphs contained in the two datasets correspond to movie collaborations. The vertices of each graph represent actors/actresses and two vertices are connected by an edge if the corresponding actors/actresses appear in the same movie. Each graph is the ego-network of an actor/actress, and the task is to predict which genre an ego-network belongs to (Yanardag & Vishwanathan, 2015). REDDIT-BINARY and REDDIT-MULTI-5K contain graphs that model the social interactions between users of Reddit. Each graph represents an online discussion thread. Specifically, each vertex corresponds to a user, and two users are connected by an edge if one of them responded to at least one of the other's comments. The task is to classify graphs into either communities or subreddits (Yanardag & Vishwanathan, 2015). COLLAB is a scientific collaboration dataset that consists of the ego-networks of several researchers from three subfields of Physics (High Energy Physics, Condensed Matter Physics and Astro Physics). The task is to determine the subfield of Physics to which the ego-network of each researcher belongs (Yanardag & Vishwanathan, 2015).

We also evaluated the proposed model on two datasets from the Open Graph Benchmark (OGB) (Hu et al., 2020), a collection of large-scale and diverse benchmark datasets for machine learning on graphs. A summary of the two datasets is given in Table 4. The `ogbg-molhiv` dataset is a molecular property prediction dataset that is adopted from the MoleculeNet (Wu et al., 2018). The dataset consists of $41,127$ molecules and corresponds to a binary classification dataset where the task is to predict whether a molecule inhibits HIV virus replication or not. The molecules in the training, validation and test sets are divided using a scaffold

Table 4: Statistics of the 2 OGB datasets that we used in our experiments.

| Dataset | ogbg-molhiv | ogbg-code2 |
|---|---|---|
| Average # vertices | 25.5 | 125.2 |
| Average # edges | 27.5 | 124.2 |
| Node features | ✓ | ✓ |
| Edge features | ✓ | ✓ |
| Directed | – | ✓ |
| # graphs | 41,127 | 452,741 |
| # tasks | 1 | 1 |
| Split scheme | Scaffold | Project |
| Split ratio | 80/10/10 | 90/5/5 |
| Task type | Binary class. | Sub-token prediction |
| Metric | ROC-AUC | F1-score |

Table 5: Values of $\lambda$ that we used in our experiments.

| | MUTAG | D&D | NCI1 | PROTEINS | ENZYMES | IMDB BINARY | IMDB MULTI | REDDIT BINARY | REDDIT MULTI-5K | COLLAB |
|---|---|---|---|---|---|---|---|---|---|---|
| $\lambda$ | $1/5$ | $1/20$ | $1/20$ | $1/30$ | $1/20$ | $1/200$ | $1/300$ | $1/500$ | $1/400$ | $1/2000$ |

splitting procedure that splits the molecules based on their two-dimensional structural frameworks. The scaffold splitting attempts to separate structurally different molecules into different subsets. The `ogbg-code2` dataset contains a large number of Abstract Syntax Trees (ASTs) that are extracted from approximately $450,000$ Python method definitions. For each method, the AST edges, the AST nodes, and the tokenized method name are retrieved. Given the body of a method represented by the AST and its node features, the task (which is known as code summarization) is to predict the sub-tokens forming the name of the method. The ASTs for the training set are obtained from GitHub projects that do not appear in the validation and test sets. We refer the reader to (Hu et al., 2020) for more details about the OGB datasets.

## F   Parameter $\lambda$

Given an input graph $G$ and a "hidden graph" $G_i$, since the "hidden graph" is trainable, the maximum vertex degree of the product graph $G_\times$ is not known beforehand. However, in case the weights of the edges of the "hidden graph" are bounded, we can compute an upper bound to that. Let $\Delta$ denote the maximum vertex degree of $G$, $c$ denote the number of vertices of the "hidden graph" $G_i$, and $b$ the maximum edge weight of the "hidden graph". Then, we have that $\Delta_\times \leq \Delta c b$, and therefore, to guarantee convergence, we need to set $\lambda \leq 1/\Delta c b$. In practice, we empirically found that even if $\lambda$ takes higher values, the geometric series converges within a small number of iterations. Table 5 shows the value of $\lambda$ that we used for each dataset.

