# OpenReview forum: "Geometric Random Walk Graph Neural Networks via Implicit Layers"
_TMLR — Withdrawn by Authors_

### Review · Reviewer_ws6M · 2022-06-05

**Summary Of Contributions:**

The authors expand on the Random Walk Graph Neural Networks work. The original RWGN looks at random walks of fixed length and here it is extended to infinite lengths using a geometric series. They then show how this can be computed efficiently.

Finally, they compare results on a variety of datasets.

**Requested Changes:**

Main issue:
- The paper explains the GRW kernel well but doesn't really explain how it is integrated into a classification model. The paper talks about "hidden graphs" many times but doesn't really explain what they are. It also doesn't go into how the graph is classified once you get k(G,G_i). I had to go into the RWNN paper to get the exact formulation which should not happen in my mind.
- When you describe the kernel you talk about initial and stopping distribution over nodes. While the initial is uniform by definition, if you run a random walk on a general graph the stopping distribution will not be uniform. Either rename the distribution or show that the stopping distribution is indeed uniform.

Other remarks:
- As the random walk kernel is a major part of this work it would be best if you gave some intuition as to why it is useful. Why does a high/low score represent similar/dissimilar graphs.
- A kernel has a very explicit mathematical definition. While the original RWK is indeed a kernel it is not obvious, at least to me, that after the modifications it is still a kernel. While this doesn't affect the work here, as you don't really depend on it being a legitimate kernel, it is important to know if the object you claim is a kernel is indeed one. This point should be addressed, if not by showing that it is indeed a kernel then by claiming that it might not be a valid kernel.
- What about graph with edge-features? Can you generalize your method to these types of graphs?

Minor remark:
- "Among others, the graph classification and graph regression..." I would mention node-based tasks as well as basic graph learning problems.


**Strengths And Weaknesses:**

Stregths:
- The Geometric extension is interesting and novel.
- Very solid experiments. Even if it wasn't always SOTA they gave good results on some. In general the experiments were properly done.

Weaknesses:
- Writing. Some important parts were missing to understand the model (without reading RWGNN paper). The paper is well written in the sense that what it explains it explains well, but a few important things are not explained at all.

---

### Review · Reviewer_Mx11 · 2022-06-15

**Summary Of Contributions:**

This work extends the Random Walk GNN [Nikolentzos & Vazirgiannis (2020)] to take into account random walks of infinite length. The core technical contribution is to use the implicit differentiation to achieve the efficient training.

**Broader Impact Concerns:**

No concern.

**Requested Changes:**

- More clear motivation on why infinite-step random walk is needed.
- Stronger empirical results compared to the original random walk GNN.
- Random walk GNN results on the OGB datasets.
- Clearer writing separating the actual contribution from the original work of Nikolentzos & Vazirgiannis (2020).

All the above needs to be addressed for the paper to be accepted, which I think is difficult.


**Strengths And Weaknesses:**

Strength:
- The use of implicit layer is appropriate.
- The paper is well-written.


Weakness:
- Motivation is weak. In the original Random Walk GNN paper, the increase random walk step sizes did not give empirical improvement, but the authors try to extend the model to use infinite step size.
- Experiments are weak. The results on the small TU graph classification datasets are rather mixed. The results on OGB datasets are also far from SoTA. In the OGB datasets, the original random walk GNN is not compared.
- Technical contribution is weak. It is a minor extension of random walk GNN without strong empirical motivation nor empirical evidence.
- Writing is confusing. A lot of content in the method section (Section 4) has been already presented in Nikolentzos & Vazirgiannis (2020). The authors did not do a good job separating their contribution from the original work that they build on.

---

### Review · Reviewer_hhJC · 2022-06-20

**Summary Of Contributions:**

The paper deals with supervised machine learning for graphs, specifically graph classification or regression. The authors leverage known results from the graph kernel literature, namely random walk kernels (RWK), to design end-to-end trainable graph neural networks (GNNs) leveraging walk information.

Specifically, they extend the work of Nikolentzos & Vazirgiannis, 2020, which compares input graphs to learnable "hidden" graphs using a finite-length RWK. Here, the authors propose to use infinite-length random walks, by leveraging known results from the graph kernel literature.

Moreover, they use known techniques from implicit differentiation to efficiently backpropagate through the resulting architecure.

Finally, the authors conduct experiments on known benchmark datasets from the kernel and GNN literature showing good results, on par with baselines.

**Broader Impact Concerns:**

Not relevant.

**Requested Changes:**

*Critical problems*
- The work of Nikolentzos & Vazirgiannis, 2020 should be described in more detail as it seems the basis of the present work
- The paper does not distinguish clearly enough between known results and newly proposed ideas. For example, from the description in Section 4 it is not clear that all the ideas are already known and have been published (Vishwanathan et al., 2010). (I am not saying there is bad intent here.)
- Further, it is not clear if Theorem 2 easily follows from the work of (Bai et al. (2019)) or if it is a truly novel contribution
- The presentation is "clumsy" at times, e.g., Equations 1 and 2 are the same. Also, Equation 3 and the one below are the same (modulo the index)
- The empirical results, Table 1, comparing the 1/2-step RWNN and the infinite one (the main contribution) are not convincing. Further, the 1/2-step RWNNs are not used on the OGB datasets,  Figure 1 (a). That is, more convincing empirical evidence is needed, showing that the infinite RWs indeed give an edge on some datasets (over the finite ones)
- Moreover, it should be made more clear what the benefits of the RW-based GNNs are compared to standard GNNs. (Note that they are not more expressive.)
- Make clear why infinite random walks are needed in the case of *finite graphs*. This seems counterintuitive. Also, discuss work on Sugiyama & Borgwardt (2015) in more detail in that context.

*Minor problems*
- Paper needs proofreading, I stumbled on various typos and grammatical issues when working through the paper.
- Bib. entries do not follow a unified format. Full surname vs. abbreviated, et cetera


**Strengths And Weaknesses:**

*Strengths*
- Well-written and easy to follow
- Discussion of literature seems reasonable
- Reasonable experimental setup/protocol

*Weaknesses*
- The paper is incremental, it is a combination of known ideas, i.e., *neither* of the following ideas is /new/
    *  comparing to learnable "hidden" graphs (Nikolentzos & Vazirgiannis, 2020)
    *  Infinite random walks (Vishwanathan et al., 2010)
    *  Implicit backpropagation for fixed-point equations (Bai et al. (2019))
- Infinite walks do not give much improvement over finite random-walk architecture in the experiments

---

### Note · Authors · 2022-07-05

I have read and agree with the venue's withdrawal policy on behalf of myself and my co-authors.